# Contract Theory-Based Incentive Mechanism for Full Duplex Cooperative NOMA with SWIPT Communication Networks

**DOI:** 10.3390/e23091161

**Published:** 2021-09-03

**Authors:** Zhenwei Zhang, Hua Qu, Jihong Zhao, Wei Wang

**Affiliations:** 1School of Electronic and Information Engineering, Xi’an Jiaotong University, Xi’an 710049, China; qhmail.com@gmail.com (H.Q.); wangwei_525@126.com (W.W.); 2School of Software Engineering, Xi’an Jiaotong University, Xi’an 710049, China; 3School of Telecommunication and Information Engineering, Xi’an University of Posts and Telecommunications, Xi’an 710061, China; xjtuxtyjy@gmail.com

**Keywords:** contract theory, cooperative NOMA, full duplex, incentive mechanism, SWIPT

## Abstract

Cooperative Non-Orthogonal Multiple Access (NOMA) with Simultaneous Wireless Information and Power Transfer (SWIPT) communication can not only effectively improve the spectrum efficiency and energy efficiency of wireless networks but also extend their coverage. An important design issue is to incentivize a full duplex (FD) relaying center user to participate in the cooperative process and achieve a win–win situation for both the base station (BS) and the center user. Some private information of the center users are hidden from the BS in the network. A contract theory-based incentive mechanism under this asymmetric information scenario is applied to incentivize the center user to join the cooperative communication to maximize the BS’s benefit utility and to guarantee the center user’s expected payoff. In this work, we propose a matching theory-based Gale–Shapley algorithm to obtain the optimal strategy with low computation complexity in the multi-user pairing scenario. Simulation results indicate that the network performance of the proposed FD cooperative NOMA and SWIPT communication is much better than the conventional NOMA communication, and the benefit utility of the BS with the stable match strategy is nearly close to the multi-user pairing scenario with complete channel state information (CSI), while the center users get the satisfied expected payoffs.

## 1. Introduction

Both Non-Orthogonal Multiple Access (NOMA) and Simultaneous Wireless Information and Power Transfer (SWIPT) have been recognized as important enabling technologies for the massive growth in data requirements for next-generation wireless communication networks. This paper investigates the coexistence of these two new, important technologies with cooperative communication and full duplex (FD), which can extend the BS coverage area, enhance the system capacity, and enormously increase the spectrum and energy efficiency of the wireless communication networks.

The concept of cooperative NOMA with SWIPT was initially proposed in [1,2], where the center user in the NOMA cluster can act as a relay to decode and forward (DF) the edge user’s data and use power splitting (PS) to harvest energy from the same received signal. FD-enabled cooperative communication was studied by [3,4,5]; this approach can massively improve the sum transmission data rate of the NOMA cluster when the self-interference (SI) at the relaying center user is well cancelled. The performance of FD cooperative relay communication with DF is much better than amplify-and-forward (AF), as verified by [6,7]. However, since the cooperating center users are inherently rational and selfish, they are not willing for their private network information to be shared with others, and thus incentivizing a user with asymmetrical information to participate in cooperative communication is still a challenging issue to be solved.

Contract theory is widely used in economics with asymmetric information to design contracts between an employer and employee [8,9]. Asymmetric information refers to the fact that the employer does not know exactly the characteristics of the employee. Using a contract theory-based incentive mechanism model, the employer can overcome the asymmetric information and efficiently incentivize the employee by offering a contract with performance requirements and corresponding rewards. Surveying contract theory, Zhang [10] demonstrated its effectiveness to design incentive mechanisms for wireless application scenarios. In the contract theory model, participants are offered well-designed rewards based on their performances, and the objective problem is to maximize the employer’s payoff or utility. The objective problem is usually formulated with the incentive compatibility constraint that the employee’s expected payoff is maximized and the individual rationality constraint that the employee’s payoff under this contract is not below its reservation payoff.

### 1.1. Related Work

Contract theory is widely used in content sharing and delivery problems in device-to-device (D2D) communication. Chen [11], introduced contract theory to provide an effective and distinctive incentive mechanism to reduce computation and communication costs in device-to-device communication, which has the advantages of a high data transmission rate, spectrum efficiency, and energy efficiency. Chen [12], proposed an incentive mechanism to optimize the operator’s expected profit from motivating content sharing D2D communication with guaranteed service quality in both complete and incomplete information scenarios. Satisfying individual rationality and incentive compatibility, the proposed greedy algorithms with low complexity obtain fast solutions for contract design based on local optimization. Yang [13] proposed a contract-based allocation mechanism to optimize the benefits of the principal sending users and the cooperative-contract users and formulate both contract parties’ utility functions; considering the social relationships, maximum benefits can be obtained under the control of the BS by a stable match algorithm with user preference. Zhang [14] used contract theory to introduce incentive mechanisms for the D2D content-sharing problem under information asymmetry, where the BS acts as an employer to offer contracts to UEs and hire them as employees to fulfill the content transmission task. The users can be efficiently rewarded according to their performance in participating in D2D communications.

Contract theory is also widely used in cooperative relay networks, such as our proposed cooperative communication method that can effectively improve the wireless spectrum efficiency and extend the coverage of these networks. Relay nodes with selfish characteristics may acquire the asymmetric network information of the location and mobility of various nodes, channel conditions, and other factors. Zhao [15] proposed a contract-theoretic model to incentivize the relay nodes with static and dynamic information to participate in cooperative communication and maximize the source’s utility under different information scenarios. By introducing a monitoring node to effectively incentivize the potential relay nodes to cooperate in the cooperative communication network, Zhao [16] investigated the various monitoring strategies of the contract-theoretic relay incentive mechanism under the dual asymmetric information scenario to achieve the better utility of the source. To capture the dynamic characteristic of the relay nodes’ cooperative information during the long-term cooperation, Zhao [17] introduced two dynamic contract mechanisms into long-term relay incentives with different relay nodes’ relay information structures to maximize the source’s expected utility. Nazari [18] applied a contract theoretic framework over decode-and-forward parallel cooperative relay wireless networks to formulate an incentive-compatible contract to stimulate relays to cooperate to improve the utility of the source node. Hasan [19] used contract theory to tackle the problem of relay selection with asymmetric information in an OFDM-based cooperative wireless system and proposed a heuristic relay selection scheme to solve the maximizing capacity nonlinear non-separable knapsack problem in each sub-carrier under a budget constraint. Liu [20], proposed a contract-based principal-agent framework for cognitive-radio-based wireless relaying networks by designing a pricing mechanism that leads to a win–win situation between the source nodes and relay nodes.

In recent years, contract theory has been studied in NOMA scenarios. Li [21] proposed a price-based power allocation scheme for a downlink NOMA system using a Stackelberg game to jointly investigate the utility maximization of the BS by pricing and users by power allocation. Tang [22] proposed a contract-based incentive mechanism together with a mobile relay selection scheme for multi-channel cooperative NOMA systems with asymmetric information, striking a good balance between performance and complexity.

Contract theory is also applied to the energy transfer or energy trading field. Li [23] used a contract approach to study the power allocation and pricing issue for SWIPT in a downlink cellular network, where a monopoly mobile network operator (MNO) gains revenue from providing both information and energy transfer for its subscribers under incentive compatibility and individual rationality constraints. Hou [24] developed incentive mechanisms under complete and asymmetric information for wireless energy trading in radio frequency energy harvesting-based Internet of Things systems, concluding that the performance with the incentive mechanism under asymmetric information is better than the existing Stackelberg game under the complete information scenario. Liu [25] proposed a contract theory based on wireless energy harvesting in order to maximize the transfer efficiency of source nodes. Liu [26] used contract theory to design an incentive mechanism and proposed an energy trading method based on energy supply point (ESP) wireless power supply.

In the above works, contract theory is used in wireless communication networks to solve problems such as the content sharing and delivery of D2D networks, cooperative relay selection, NOMA networks, and energy transfer/trading networks. This paper uses a contract theory incentive for a relaying center user in FD cooperative NOMA and SWIPT communication networks.

### 1.2. Our Contribution

Fundamentally, the proposed FD cooperative NOMA and SWIPT communication method uses energy transfer to incentivize the relaying center user to participate in the cooperative process in an intrinsic way. Motivated by the above findings, the aim of this paper is to design incentive mechanisms with contract theory under the asymmetric information situation to incentivize the relaying center user to participate in FD cooperative NOMA and SWIPT communication networks so that the BS can obtain the maximum benefit utility while guaranteeing the edge user and center user’s transmission rates, and the center user can obtain extra energy transmissions as the compensation for participate in the cooperative communication. The contributions of this work are fourfold:In FD cooperative NOMA and SWIPT communication networks, we design a contract-theory based incentive mechanism to encourage central users to participate in cooperative transmission. The type of the contract is determined by the channel state information of the relay user’s two transmission stages.Under the asymmetric information scenario, based on the adverse selection of a contract theory-based incentive mechanism, the BS obtains a contract price (sub-channel transmission power) in the complete channel information, which is regarded as the upper bound of the asymmetric information scenario. The asymmetric channel information is estimated by the BS to design a contract to achieve the maximum benefit utility.Receiving the contract from the BS, the center users with asymmetric information adopt the optimization method of power allocation to evaluate the extra energy transmission from the BS and give feedback to the BS to confirm the execution of the contract.In the multi-user pairing scenario, the preference lists of the center user and the edge user participating in cooperative transmission are established. The BS uses the stable match strategy [27,28] to design the contracts to achieve the maximum stable benefits while satisfying the expected payoffs of each center user.

The organization of this paper as follows: Section 1 introduces FD cooperative NOMA and SWIPT communication, contract theory in wireless communication networks, and the contribution of this paper; Section 2 gives the system model; Section 3 designs the contract between BS and the center user; Section 4 formulates the objective problem; Section 5 presents the incentive mechanism in complete and incomplete channel information scenarios; Section 6 proposes the stable matching algorithm; Section 7 discusses the simulations in four aspects; and Section 8 concludes this work.

## 2. System Model

We consider a multi-user pairing cooperating communication for sub-6G networks with one BS, coverage with *M* edge users and *N* center users (that can work with SWIPT and as a relay in FD cooperative communication), and *K* (K=M=N) sub-channels. Each FD downlink cooperative NOMA with SWIPT communication consists of one center user and one edge user on one sub-channel. Due to the strong obstruction (large-scale path fading) between the BS and the edge user, the BS needs to incentivize a center user as a relay to engage in cooperative communication with the edge user in NOMA to satisfy the data rate requirement. We assume that each relaying center user is provided with one receive antenna and one transmission antenna and works in FD mode with perfect self-interference cancellation (SIC).

The BS transmits the information data to the cooperative communication users in the *k*th sub-channel using superposition coding (SC) technology. The relaying center user with a good channel condition receives the signal from the BS and uses power splitting (PS) technology in SWIPT with β for energy harvesting and 1−β for information decoding. Concurrently, the user uses SIC technology to subtract the edge user information and acts as a relay to decode-and-forward it. The edge user receives the signal from the BS and the cooperative relaying center user and then decodes the information in the maximal-ratio combining (MRC) mode. A FD cooperative NOMA and SWIPT communication network consists of two transmission stages, which are detailed below in Figure 1 and Figure 2.

### 2.1. Direct Transmission Stage

The observation for the relaying center user *n* is given by [1]
(1)$$y_{sn}^{k}=\sum_{i=\left\{n,m\right\}}\sqrt{p_{i}^{k}\left(1-\beta_{nm}^{k}\right)}H_{sn}^{k}x_{i}^{k}+\sqrt{p_{rn}^{k}}H_{rn}^{k}x_{m}^{k}\frac{\left(T-\tau \right)}{T}+n_{n}^{k}$$
where *k* means that the BS’s transmission to the cooperative users is in the *k*th sub-channel, $p_{i}^{k}$ is the transmit power allocated to users in the *k*th sub-channel, $H_{sn}^{k}$ models the channel gain from the BS to the relaying center user *n*, $\beta_{nm}^{k}$ is the center user *n*’s energy transmission coefficient, $n_{n}^{k}$ is the additive white Gaussian noise (AWGN) with variance $\sigma^{2}$, and $H_{rn}^{k}$ models the self-interference fading channel gain. We assume ${\left|p_{n}^{k}\right|}^{2}+{\left|p_{m}^{k}\right|}^{2}=p^{k}$ and ${\left|p_{n}^{k}\right|}^{2}=p^{k}\omega_{nm}^{k}$, ${\left|p_{m}^{k}\right|}^{2}=p^{k}\left(1-\omega_{nm}^{k}\right)$, where $p^{k}$ is the allocated power on the *k*th sub-channel. $\omega_{nm}^{k}$ is the edge user *m*’s power allocation coefficient. $x_{n}$ and $x_{m}$ are the transmission signals for the relaying center user and the edge user. $p_{rn}^{k}$ is the transmit power for relaying a transmission to the edge user *m*. τ is the time delay for the relaying center user caused by FD operation, which can be negligible owing to the τ being far smaller than the time slot *T* [29,30].

The received SINR at relaying center user *n* to detect signal $x_{m}$ from edge user *m* is given by
(2)$$\gamma_{sm,n}^{k}=\frac{p^{k}H_{sn}^{k}\omega_{nm}^{k}\left(1-\beta_{nm}^{k}\right)}{p^{k}H_{sn}^{k}\left(1-\omega_{nm}^{k}\right)\left(1-\beta_{nm}^{k}\right)+\kappa p_{rn}^{k}H_{{}_{rn}}^{k}+\sigma^{2}}$$

We assume that the FD self-inference cancellation technology plays an important role; the residual self-interference cancellation coefficient κ is smaller than −50 dB, and $H_{rn}^{k}$ is about −10 dB, so $\kappa p_{rn}^{k}H_{rn}^{k}$ is much smaller than noise and can be neglected. Meanwhile, the power consumption of the SIC process is ignored for linear energy harvesting communication; otherwise, non-linear energy harvesting would need to be considered, such as in [6].

The received SINR at relaying center user *n* to detect signal $x_{n}$ of center user *n* is given by:(3)$$\gamma_{sn}^{k}=\frac{p^{k}H_{sn}^{k}\left(1-\omega_{nm}^{k}\right)\left(1-\beta_{nm}^{k}\right)}{\kappa p_{rn}^{k}H_{{}_{rn}}^{k}+\sigma^{2}}$$For relaying center user *n*, we assume that the capacity of the battery is large enough and the energy harvesting is linear with energy conversion coefficient η which is from 0∼1. In each time slot T=1, the harvested energy is represented by
(4)$$E_{n}^{k}=\eta \beta_{nm}^{k}p^{k}H_{sn}^{k}$$

### 2.2. Cooperative Transmission Stage

The observation for edge user *m* is given by
(5)$$y_{sm}^{k}=\sum_{i=\left\{n,m\right\}}\sqrt{p_{i}^{k}\left(1-\beta_{nm}^{k}\right)}H_{sm}^{k}x_{i}^{k}+\sqrt{p_{rn}^{k}}x_{m}^{k}G_{nm}^{k}\frac{\left(T-\tau \right)}{T}+n_{m}^{k}$$
where $H_{sm}$ models the channel gain from the BS to the edge user *m*, $G_{nm}$ models the channel gain from the center user *n* to the edge user *m*. $n_{m}^{k}$ is the AWGN with variance $\sigma^{2}$.

Using MRC, edge user *m* combines the signals from the BS and the cooperative relaying center user *n*, and the SINR for edge user *m* to detect $x_{m}$ is as follows:(6)$$\gamma_{sm,MRC}^{k}=\frac{p^{k}H_{sm}^{k}\omega_{nm}^{k}}{p^{k}H_{sm}^{k}\left(1-\omega_{nm}^{k}\right)+\sigma^{2}}+\frac{p_{rn}^{k}G_{nm}^{k}}{\sigma^{2}}$$

## 3. Contract Formulation

A contract is composed of the BS transmission power and the contract type with the cooperative communication requirement. The whole process of the contract formulation consists of three phases [15]—service requesting and CSI interaction, contract design and confirmation, and contract execution—as described in Figure 3.

### 3.1. BS’s Benefit Utility

Phase 1: *service requesting and CSI interaction.* An edge user and a center user request data transmission services with minimum data requirements at the same time; then, the BS requests the CSI between the users and BS.

Phase 2: *contract design and confirmation.* The BS designs and offers the contract $\Phi_{nm}^{k}$ designed by the contract theory-based incentive mechanism to maximize its benefits while satisfying the center user’s expected payoffs in the FD cooperative NOMA and SWIPT communication process. After receiving the contract with the complete channel state information, the relaying center user optimizes the coefficients of the FD cooperative NOMA with SWIPT communication process to satisfy the data rate requirements of both cooperative users and obtain the maximum payoffs. The user will decide whether to accept or drop the contract.

Phase 3: *contract execution.* If the relaying center user accepts the contract, it replies to the BS with the optimal cooperative transmission coefficients. Then, the BS executes the contract to meet the data rate requirements of the edge user and the center user and obtains rewards from the users. The relaying center user obtains extra transmission energy from the radio frequency signals sendt by the BS in the cooperative transmission process as a payoff.

Either an edge user or a center user would receive a fixed data rate service from the BS and pay the corresponding reward for it. The BS is well aware of all CSI between itself and the users, but it has no knowledge of the CSI between the center user and the edge user. The aim of the BS is to incentivize the center user to participate in the cooperative transmission with the edge user to decrease the transmission power and obtain the maximum network benefit utility. It will offer a contract set to incentivize the center user to engage in cooperative communication to achieve the data rate requirements of the edge users. The BS estimates the full CSI between the center user and edge user to design a contract with the maximum benefit for itself and minimum payoffs for the center user as soon as possible. The BS’s benefit utility can be defined as
(7)$$U_{BS}^{k}=\lambda \cdot \left(R_{n}^{k}+R_{m}^{k}\right)-c\cdot p^{k}$$
where $R_{n}^{k}$ and $R_{m}^{k}$ are the transmission date rate requirements of the center user and edge user in the *k*th channel, respectively. Without loss of generality, λ is the unit price of the data rate requirement, and *c* is the unit cost of the BS transmission power.

### 3.2. Payoff for Relaying Center User

Owing to the rational and selfish nature of the relaying center users, once the center user receives the contract, it would optimize the coefficients $\left(\omega_{nm}^{k},\beta_{nm}^{k}\right)$ of the cooperative communication to maximize its own utility and then reply to the BS to accept or drop the contract. The utility of the relaying user is
(8)$$U_{nm}^{k}=c\cdot \left(E_{n}^{k}-P_{rn}^{k}T\right)=c\cdot \left(\eta \beta_{nm}^{k}p^{k}H_{sn}^{k}-P_{rn}^{k}\right)$$
where $E_{n}^{k}=T\eta \beta_{nm}^{k}p^{k}H_{sn}^{K}$ is the harvested energy of center user *n* in the *k*th subchannel using PS in SWIPT, and $P_{rn}^{k}$ is easy to obtain with the given contract after achieving the optimal coefficients $\left(\omega_{nm}^{k},\beta_{nm}^{k}\right)$.

### 3.3. Optimal Coefficients for Each Given Contract

The decoded data rate of each part in the FD cooperative NOMA and SWIPT communication network is given as
(9)$$\left\{\begin{array}{c} R_{sm,n}=\log\left(1+\frac{p^{k}H_{sn}^{k}\cdot \omega_{nm}^{k}\left(1-\beta_{nm}^{k}\right)}{p^{k}H_{sn}^{k}\left(1-\omega_{nm}^{k}\right)\left(1-\beta_{nm}^{k}\right)+\sigma^{2}+\kappa p_{rn}^{k}H_{{}_{rn}}^{k}}\right)\\ R_{sn}=\log\left(1+\frac{p^{k}H_{sn}^{k}\left(1-\omega_{nm}^{k}\right)\left(1-\beta_{nm}^{k}\right)}{\sigma^{2}+\kappa p_{rn}^{k}H_{{}_{rn}}^{k}}\right)\\ R_{sm,MRC}=\log\left(1+\frac{p^{k}H_{sm}^{k}\cdot \omega_{nm}^{k}}{p^{k}H_{sm}^{k}\left(1-\omega_{nm}^{k}\right)+\sigma^{2}}+\frac{p_{rn}^{k}G_{nm}^{k}}{\sigma^{2}}\right) \end{array}\right.$$

The minimum data rate requirements for the edge user and center user are $R_{m}^{\min}$ and $R_{n}^{\min}$.

**Characterization** **1.**
*The maximum payoffs of a relaying center user when it receives a contract are obtained with the optimization power allocation coefficient $\left(\omega_{nm}^{k},\beta_{nm}^{k}\right)$ and the Karsh–Kuhn–Tucker point, where $R_{sm,MRC}=R_{m}^{\min}$, $R_{sm,n}=R_{sn}=R_{n}^{\min}$.*


**Proof.** The proof can be found in Appendix A.    □

According to Characterization 1, let $\gamma_{m}=2^{R_{m}^{\min}}-1$ and $\gamma_{n}=2^{R_{n}^{\min}}-1$; the optimal coefficients can be obtained when $P^{k}\geq \frac{\left(\gamma_{m}\gamma_{n}+\gamma_{m}+\gamma_{n}\right)\sigma^{2}}{H_{\mathrm{sn}}^{k}}$ is satisfied:(10)ωnmk*=γmγn+1γmγn+1γmγn+γm+γnγmγn+γm+γn
(11)βnmk*=1−γmγn+γm+γnσ2γmγn+γm+γnσ2pkHsnkpkHsnk

### 3.4. Contract Type

The payoffs of the *n*th relaying center user from the cooperative communication with the *m*th edge user are redefined as
(12)$$U_{nm}^{k}=c\cdot \left(E_{n}^{k}-P_{rn}^{k}T\right)=c\cdot \eta \beta_{nm}^{k}H_{sn}^{k}\left(p^{k}-\Pi_{m}^{k}\cdot \theta_{m}^{k}\right)$$
(13)$$\Pi_{m}^{k}=\gamma_{m}-\frac{p^{k}H_{sm}^{k}\omega_{nm}^{k}}{p^{k}H_{sm}^{k}\left(1-\omega_{nm}^{k}\right)+\sigma^{2}}$$

We define the contract type of relaying center user *n* in cooperative communication with edge user *m* as
(14)θmk=σ2σ2ηHsnkGnmkηHsnkGnmk.
which suggests that the stronger the multipliers of the channel gains from the BS to the relaying center user and from the relaying center user to the edge user, the lower the contract type of $\theta_{m}^{k}$. Without loss of generality, we assume that there are *M* types of edge users in total cooperatively communicating with one relaying center user *n* in the *k*th subchannel. We denote the set of types as $S=\left(\theta_{1}^{k},\theta_{2}^{k},\ldots ,\theta_{M}^{k}\right)$ with $\left|S\right|=M$ and $\theta_{1}^{k}\leq \theta_{2}^{k}\leq \ldots \leq \theta_{M}^{k}$.

### 3.5. IR and IC Constraints of Contract

In sub-6G networks, one relaying center user would receive *M* types of contracts for cooperative communication with *M* different edge users. Since the relaying center users are rational and selfish, the designed contract $\Phi_{nm}^{k}=\left(p_{}^{k},\theta_{m}^{k}\right)$ should be feasible and satisfy the following individual rationality (IR) and incentive compatibility (IC) constraints [14]. The IR condition requires that the received energy of each relaying center user should compensate the cost when it participates in the cooperative communication. The IC condition ensures that the relaying center user automatically selects the contract item that maximizes the relaying center user’s payoffs. If a contract satisfies the IR and IC constraints, the contract is referred to as a feasible contract.

**Definition** **1**(**IR constraints**). *A contract item for which a relaying center user signs for its type should ensure a non-negative payoff:*
(15)$$U_{nm}^{k}=c\cdot \eta \beta_{nm}^{k}H_{sn}^{k}\left(p_{}^{k}-\Pi_{m}^{k}\theta_{m}^{k}\right)\geq 0,\forall \theta_{m}^{k}\in S$$

**Definition** **2**(**IC constraints**). *The payoff of the relaying center user is maximized when signing for its type.*
(16)$$\left(p_{m}^{k}-\Pi_{m}^{k}\theta_{m}^{k}\right)\geq \left(p_{m^{\prime }}^{k}-\Pi_{m^{\prime }}^{k}\theta_{m^{\prime }}^{k}\right),\forall \theta_{m}^{k},\theta_{m^{\prime }}^{k}\in S,m<m^{\prime }$$

## 4. Problem Formulation

The aim of this work is to find feasible contract sets that maximize the expected benefit utility of the BS. The problem is formulated as **P1**,
(17)$$\begin{array}{c} \max U_{BS}=\sum_{k=1}^{K}\sum_{n=1}^{N}\sum_{m=1}^{M}\alpha_{nm}^{k}\left(\lambda \cdot \left(R_{n}^{k}+R_{m}^{k}\right)-c\cdot P^{k}\right),\\ s.t.\max U_{nm}^{k}=c\eta \beta_{nm}^{k}H_{sn}^{k}\left(p_{n}^{k}-\Pi_{n}^{k}\theta_{n}^{k}\right),\\ IC:\left(p_{m}^{k}-\Pi_{m}^{k}\theta_{m}^{k}\right)\geq \left(p_{m^{\prime }}^{k}-\Pi_{m^{\prime }}^{k}\theta_{m^{\prime }}^{k}\right),\\ IR:U_{nm}^{k}=c\cdot \eta \beta_{nm}^{k}H_{sn}^{k}\left(p_{m}^{k}-\Pi_{m}^{k}\theta_{m}^{k}\right)\geq 0,\\ SINR:R_{n}^{n}\geq R_{n}^{\min},R_{mn}^{n}\geq R_{m}^{\min},R_{m}^{mrc}\geq R_{m}^{\min}. \end{array}$$

**P1** is a Mixed-Integer Non-Linear Programming (MINLP) problem, where $\alpha_{nm}^{k}=1$ is the index of edge user *m* and relaying center user *n* which are in cooperative communication in the *k*th sub-channel. The problem is divided into two power allocation and user pairing sub-problems. In the power allocation sub-problem, a contract theory-based incentive mechanism is adopted to maximize the BS’s benefit utility while maximizing the relaying center users’ payoffs. In the multi-user pairing sub-problem, a match theory with the Gale–Shapley (GS) algorithm is proposed, and the computational complexity is much lower than that of the global searching algorithm.

## 5. Incentive Mechanism for Power Allocation of the Contract

For the power allocation sub-problem in a single contact design, problem **P1** can be transformed into problem **P2**,
(18)$$\begin{array}{c} \max U_{}^{k}=\lambda \cdot \left(R_{n}^{k}+R_{m}^{k}\right)-c\cdot \left(P^{k}-P_{N}^{noma}\right)\\ s.t.\max U_{nm}^{k}=c\cdot \eta \beta_{nm}^{k}H_{sn}^{k}\left(p_{m}^{k}-\Pi_{m}^{k}\theta_{m}^{k}\right),IC,IR,SINR. \end{array}$$
where $P_{N}^{noma}$ is the fraction transmission power used to satisfy the QoS of center user *n*, which is fixed in the conventional NOMA and the proposed FD cooperative NOMA and SWIPT communication networks, and with $P_{N}^{noma}$, it is easy to obtain that $P_{N}^{noma}=\gamma_{n}\sigma^{2}/H_{sn}^{k}$.

### 5.1. Complete Channel Information Scenarios

We consider the complete information scenario as the upper-bound of the optimization.

**Lemma** **1.***For the optimal feasible contract set $\Phi_{nm}^{k}=\left(p_{m}^{k},\theta_{m}^{k}\right)$ with the complete information scenario, the IR constraints in Equation (Equation 8) are equivalent to $U_{nm}^{k}=0$*.

We assume the optimal contract set $\Phi_{nm}^{k}=\left(p_{}^{k},\theta_{m}^{k}\right)$ with power as
(19)pmk=−αn1+αn12+4γnHsnkHsmkαn2−αn1+αn12+4γnHsnkHsmkαn22γnHsnkHsmk2γnHsnkHsmk
where $\Delta =\left(\gamma_{m}\gamma_{n}+\gamma_{m}+\gamma_{n}\right)$, $\alpha_{m}^{1}=\Delta \cdot \sigma^{2}\left(H_{sn}^{k}-\gamma_{n}H_{sm}^{k}\right)-\gamma_{m}\theta_{m}^{k}H_{sn}^{k}H_{sm}^{k}$, and $\alpha_{m}^{2}=\Delta \cdot \sigma^{2}\left(\theta_{m}^{k}\gamma_{m}H_{sn}^{k}-\Delta \cdot \sigma^{2}\right)$.

### 5.2. Incomplete Channel Information Scenarios

**Lemma** **2.**
*For the optimal feasible contract set $\Phi_{nm}^{k}=\left(p_{}^{k},\theta_{m}^{k}\right)$, the following condition holds: $p_{m}<p_{m^{\prime }}$ if and only if $\theta_{m}<\theta_{m^{\prime }}$.*

(20)
$$P_{1}^{k}\leq P_{2}^{k}\leq \ldots \leq P_{m}^{k};\theta_{1}^{k}\leq \theta_{2}^{k}\leq \ldots \leq \theta_{m}^{k}$$



**Lemma** **3.**(***IC Transitivity***) *For the optimal feasible contract set, suppose the condition in Lemma 2 holds, and the IC constraints are equivalent to*
(21)$$p_{1}^{k}-\Pi_{1}^{k}\theta_{1}^{k}\geq p_{2}^{k}-\Pi_{2}^{k}\theta_{2}^{k}\geq \ldots \geq p_{}^{k}-\Pi_{M}^{k}\theta_{M}^{k}$$

**Theorem** **1.**
*The contract $\Phi_{nm}^{k}=\left(p_{nm}^{k},\theta_{m}^{k}\right)$ is feasible when the following conditions hold:*

(22)
$$\begin{array}{c} \left(1\right)\;P_{1}^{k}\leq P_{2}^{k}\leq \ldots \leq P_{M}^{k}\\ \left(2\right)\;m=M,p_{m}^{k}-\Pi_{m}^{k}\theta_{m}^{k}=0\\ \left(3\right)\;m\leq M,p_{m-1}^{k}-\Pi_{m-1}^{k}\theta_{m-1}^{k}\geq p_{m}^{k}-\Pi_{m}^{k}\theta_{m}^{k}=U_{nm}^{k} \end{array}$$



To optimize **P2**, the unique and optimal power allocation is
(23)$$p_{M}^{k}=\frac{-\alpha_{m}^{1}+\sqrt{{\left(\alpha_{m}^{1}\right)}^{2}+4\gamma_{n}H_{sn}^{k}H_{sm}^{k}\alpha_{n}^{2}}}{2\gamma_{n}H_{sn}^{k}H_{sm}^{k}}$$
(24)$$p_{m-1}^{k}=\frac{-\alpha_{m-1}^{1}+\sqrt{{\left(\alpha_{m-1}^{1}\right)}^{2}+4\gamma_{n}H_{sn}^{k}H_{sm}^{k}\alpha_{m-1}^{2}}}{2\gamma_{n}H_{sn}^{k}H_{sm}^{k}}$$
where $\Delta =\gamma_{m}\gamma_{n}+\gamma_{m}+\gamma_{n}$, αm−11=Δσ2Hsnk−γnHsmk+θm−1kHsmkUnmkUnmkηη−γmHsnk,

αm−12=Δσ2θm−1kγmHsnk+UnmkUnmkη−Δσ2η−Δσ2, and $U_{nm}^{k}$ is get from Equation (Equation 11).

The details of the incentive mechanism for power allocation are presented in Algorithm 1.
**Algorithm 1** Power allocation optimization algorithm of center users in complete/incomplete information scenarios.**Input:** Contract $\Phi_{nm}^{k}=\left(p^{k},\theta_{m}^{k}\right)$ accepted from BS to cooperative transmission to edge user *m*. The required data rate of edge user $R_{m}^{\min}$ and center user $R_{n}^{\min}$.**Output:** Accepted or Not ? Cooperative transmission coefficient $\left(\omega_{mn}^{k},\beta_{mn}^{k}\right)$.Let $\gamma_{m}=2^{R_{m}^{\min}}-1$ and $\gamma_{n}=2^{R_{n}^{\min}}-1$, and $\Delta =\left(\gamma_{m}\gamma_{n}+\gamma_{m}+\gamma_{n}\right)$, get the upper-bound and lower-bound cooperative transmission power by: $P_{up}^{k}=\Delta \sigma^{2}/H_{sm}^{k},P_{low}^{k}=\Delta \sigma^{2}/H_{sn}^{k}.$**if**$P^{k}\leq P_{\mathrm{up}}^{k}\;and\;P^{k}\geq P_{\mathrm{low}}^{k}.$**then**Get the optimal edge user power coefficient by$$\omega_{nm}^{k*}=\gamma_{m}\left(\gamma_{n}+1\right)/\left(\gamma_{m}\gamma_{n}+\gamma_{m}+\gamma_{n}\right)$$Get the optimal center user’s energy transmission coefficient by
$$\beta_{nm}^{k*}=1-\left(\gamma_{m}\gamma_{n}+\gamma_{m}+\gamma_{n}\right)\sigma^{2}/\left(p^{k}H_{sn}^{k}\right)$$**In complete scenarios:**Get the optimal power transmission for FD cooperative NOMA and SWIPT communication by
$$p_{m}^{k}=\frac{-\alpha_{m}^{1}+\sqrt{{\left(\alpha_{m}^{1}\right)}^{2}+4\gamma_{n}H_{sn}^{k}H_{sm}^{k}\alpha_{m}^{2}}}{2\gamma_{n}H_{sn}^{k}H_{sm}^{k}}$$
where $\alpha_{m}^{1}=\Delta \cdot \sigma^{2}\left(H_{sn}^{k}-\gamma_{n}H_{sm}^{k}\right)-\gamma_{m}\theta_{m}^{k}H_{sn}^{k}H_{sm}^{k}$, $\alpha_{m}^{2}=\Delta \cdot \sigma^{2}\left(\theta_{m}^{k}\gamma_{m}H_{sn}^{k}-\Delta \cdot \sigma^{2}\right)$.**In incomplete scenarios:**Get the optimal power transmission for the *n*th type contract relaying center user by
$$p_{M}^{k}=\frac{-\alpha_{m}^{1}+\sqrt{{\left(\alpha_{m}^{1}\right)}^{2}+4\gamma_{n}H_{sn}^{k}H_{sm}^{k}\alpha_{m}^{2}}}{2\gamma_{n}H_{sn}^{k}H_{sm}^{k}}$$**for**m≥2**do**Get the optimal power transmission for the *n*th type contract relaying user through recursion by
$$p_{m-1}^{k}=\frac{-\alpha_{m-1}^{1}+\sqrt{{\left(\alpha_{m-1}^{1}\right)}^{2}+4\gamma_{n}H_{sn}^{k}H_{sm}^{k}\alpha_{m-1}^{2}}}{2\gamma_{n}H_{sn}^{k}H_{sm}^{k}}$$
where αm−11=Δσ2Hsnk−γnHsmk+θm−1kHsmkUnmkUnmkηη−γmHsnk, αm−12=Δσ2θm−1kγmHsnk+UnmkUnmkη−Δσ2η−Δσ2, $U_{nm}^{k}=c\cdot \left(\eta \beta_{nm}^{k}p^{k}H_{sn}^{k}-P_{rn}^{k}\right)$.then m=m−1;**end for****return** Accept the contract with the transmission power $p_{}^{k}$ and coefficient $\left(\omega_{mn}^{k*},\beta_{mn}^{k*}\right)$;**else**The BS only can use NOMA transmission mode to satisfy the QoS of both edge users and center users.**end if**

## 6. Stable Match for Multi-User Pairing Scenarios

As mentioned in the proposed system model in Section 2, there may be multiple edge users simultaneously competing for the same cooperative relaying center user in the networks, and the same center user may be selected as the optimal relaying user by multiple edge users. This competition conflict will cause contract delivery to fail. Therefore, we introduce the user preference for the relaying center user’s adverse selection on the basis of match theory with the Gale–Shapley algorithm [31,32], which mainly emphasizes the match stability and is usually used to solve the stable marriage problem between men and women.

Based on the contract theory incentive mechanism, the BS applies a feasible contract set to the relaying center user for each edge user, So, each center user would receive *M* contracts for cooperative communication with *m* different edge users. With the proposed Algorithm 1, the user would obtain the feasible contract set with the center user *n*’s preference list in descending order of recursion.

At the BS side, the BS calculates the benefit utility $U_{m}$ for cooperative communication with edge user *m* and can obtain the edge user *m*’s preference list in descending order of recursion.
(25)$$U_{m}=\lambda \cdot R_{m}^{k}-c\cdot \left(P^{k}-{P_{N}}^{noma}\right)$$

After several rounds of matches, conflict can be avoided and a stable match for optimal contract delivery can be obtained. The details of the GS match algorithm are summarized as in Algorithm 2 and Figure 4.
**Algorithm 2** Gale and Shapley stable match algorithm with preference.Set up cooperative relaying center user’s preference list PLc; $\forall n_{k}\in N$;Set up edge users’ preference list PLe; $\forall m_{k}\in M$;Set up a list of unmatched edge users $UM=\left\{m_{k},\forall m_{k}\in M\right\}$;**while**UM is not empty **do**$\;\;m_{k}$ proposes to the cooperating relaying center user $n_{k}$ located first in its list, $\forall m_{k}\in UM$  **if**
$n_{k}$ receives a proposal from $m_{k^{\prime }}$, and $m_{k^{\prime }}$ is more preferred than the current hold $m_{k}$ **then**$\;\;\;n_{k}$ holds $m_{k^{\prime }}$ and rejects $m_{k}$;$\;\;\;m_{k^{\prime }}$ is removed from UM and $m_{k}$ is added into UM;**else**$\;\;\;n_{k}$ rejects $m_{k^{\prime }}$ and continues holding $m_{k}$;  **end if****end while**

The computational complexity of the stable match with the GS algorithm is $N^{2}$ in the worst case, Nlog(N) in the average case, and *N* in the best case, which is much more efficient than the global search method.

## 7. Discussion

In the simulations, we assumed the relaying center users and edge users were randomly distributed and deployed in the area of 20∼50 and 50∼100 m Euclidean distance to the BS, respectively. The distance-dependent pass loss model followed [1]. The detail of the coefficients are given in Table 1. All the simulation results were averaged over 1000 independent channel realizations, and the performance analysis results of the proposed FD cooperative NOMA and SWIPT transmission approach were compared with conventional NOMA transmission [33,34].

### 7.1. Discussion of the FD Cooperative NOMA and SWIPT Transmission Mode in Complete Channel Information Scenario

Figure 5 presents the network transmission performance of FD NOMA and SWIPT cooperative transmission in sub-6G networks with different transmission coefficients. Figure 5a presents the relation of three decoded transmission rates with the edge user’s power allocation coefficient ω. The transmission power of the subchannel was set at 30 dBm (1 W), the distances of the center user and edge user to the BS were 20 m and 80 m, their deviation angle was $\delta =30^{\circ }$, the center user’s energy transmission coefficients β were set as 0.3, 0.6, and 0.9, respectively, the center user’s energy conversion efficiency η was set as 0.9, and we neglected the impact of FD self-interference.

Figure 5a indicates that, with the increase of the edge user’s power allocation coefficient, the center user’s decoded transmission rate monotonically decreases, the rate at which the center user decodes the edge user’s transmission rate slowly monotonically increases, and the edge user’s decoded transmission rate monotonically increases Figure 5b presents the relation of the three decoded rates with the energy transmission coefficient β. The edge user’s power allocation coefficients were set as 0.3, 0.6, and 0.9, respectively, while other coefficients were set to the same values as shown in Figure 5a. Figure 5b indicates that, with the increase of the center user’s energy transmission coefficient, the center user’s decoded transmission rate monotonically decreases, the rate at which the center user decodes the edge user’s transmission rate slowly monotonically decreases, and the edge user’s decoded transmission rate monotonically increases. Figure 5c presents the relation of the three decoded rates with the center user’s energy conversion efficiency η. The coefficient pair (ω,β) was set as (0.4, 0.6), (0.2, 0.8), and (0.5, 0.5), respectively. Figure 5c indicates that the center user’s energy conversion efficiency only positively increases with the increase of the edge user’s decoded transmission rate. Figure 5d presents the relation of three decoded data rates with the FD self-interference cancellation residual coefficient κ. The coefficient (ω,β) pair was set as (0.2, 0.8), (0.5, 0.8), and (0.8, 0.8), respectively. Figure 5d indicates that, with the increase of the FD self-interference cancellation residual coefficient, the edge user’s decoded transmission rate did not change, the rate at which the center user decodes the edge user’s transmission rate slowly monotonically decreases, and the center user’s decoded transmission rate monotonically decreases.

Figure 6 presents the relation of the maximum achievable rate of the edge user with the transmission power in four groups of cooperative transmission matched user pairings. We compare the edge user’s achievable transmission rate with the proposed FD cooperative NOMA and SWIPT transmission, conventional NOMA transmission, and OMA transmission with the transmission power increasing from 10 dBm to 46 dBm. Figure 6a presents the scenario in which the distances from the center user and the edge user to the BS were set as 20 and 80 m, and the deviation angle was set as $\delta =30^{\circ }$. Figure 6b presents the corresponding scenario for (20,80), $\delta =30^{\circ }$; Figure 6c discussed the corresponding scenario is (20,80), $\delta =60^{\circ }$; Figure 6d presents the corresponding scenario for (20,50), $\delta =60^{\circ }$.

Figure 6 indicates that, in four group scenarios, the simulation result (SIM) is approximately equal to the theoretical analysis result (ANLS), with an error less than 0.2%. With the increase of transmission power, under the different scenarios and different transmission modes, the edge user’s transmission rate monotonically increases. When the transmission power is 25 dBm, for example in Figure 6a, the maximum edge user rate is 8.2 Mbps in the FD cooperative NOMA and SWIPT transmission mode, which is 127% better than that in NOMA mode with 3.6 Mbps and 147% better than that in OMA mode with 3.3 Mbps. Comparing the four group scenarios, the location distribution of the center user and the edge user have a great influence on the edge user’s achievable maximum transmission rate in the FD cooperative NOMA and SWIPT transmission mode. The transmission rates in the conventional NOMA mode and OMA mode are only related to the CSI between the user and BS.

### 7.2. Contract Type in Complete Channel Information Scenario

Figure 7 shows the histogram of the contract type value with the change of the center user and the edge user’s location distributions according to Equation (Equation 7). The distances between the center user and the BS and between the edge user and the BS were fixed under three different user pair scenarios, Figure 7a indicates that the contract type value increases with the decrease of the deviation angles between the center user, the edge user, and the BS, which satisfies the IC constraint in Equation (Equation 10).

Figure 7b–d separately set the deviation angles between the center user, the edge user, and the BS as $\delta =30^{\circ }$, 45${}^{\circ }$, and $60^{\circ }$ under the three scenarios, with the distances between the center user and the BS set as 20, 30, and 40 m. The simulation results indicate that the contract type value increases with the decrease of the distance between the edge user and the BS, which also satisfies the IC constraint in Equation (Equation 10).

The transmission power was set as 1 W (30 dBm) and 2 W (33 dBm), respectively, in the contract. Under different transmission rate requirements of the center user and energy conversion efficiency, and under different center user and edge user location distributions, Figure 8 presents the center user’s utility (payoffs) versus the edge user’s transmission rate requirement. Figure 8a corresponds to the scenario with $R_{n}^{min}$ = 2 Mbps, η=0.9, Figure 8b corresponds to the scenario with $R_{n}^{min}$ = 2 Mbps, η=0.6, Figure 8c corresponds to the scenario with $R_{n}^{min}$ = 5 Mbps, η=0.9, and Figure 8d corresponds to the scenario with $R_{n}^{min}$ = 3 Mbps, η=0.6. Figure 8 indicates that, with the increase of the edge user’s transmission rate requirement $R_{m}^{min}$, the center user’s utility monotonically decreases. After a threshed point, the center user’s utility decreases to 0, which means that the full duplex cooperative NOMA and SWIPT transmission mode cannot satisfy the demands of the user’s data rate. Comparing the four group scenarios, this indicates that a higher transmission rate requirement for the center user would significantly decrease its utility, and the energy transfer efficiency would also significantly affect its utility.

We set the center user transmission rate requirement as $R_{n}^{min}$ and the energy conversion efficiency as η=0.9 and η=0.7, respectively. Figure 9 presents the BS utility (benefit) versus the edge user transmission rate requirement, comparing three groups using FD NOMA and SWIPT cooperative transmission with one group using conventional NOMA transmission in a complete channel state information scenario. Figure 9 indicates that, when the edge user’s data rate requirement $R_{n}^{min}$ is low, the BS’s utility in conventional NOMA transmission mode is better than the proposed FD cooperative NOMA and SWIPT transmission, with the utility being 0. When increasing $R_{m}^{min}$, the conventional NOMA transmission mode cannot satisfy the user’s date rate requirement, and the proposed cooperative transmission mode can achieve increased utility until a threshold. Then, the BS utility in the proposed cooperative transmission mode decreases to 0, which means that it cannot satisfy the $R_{n}^{min}$ = 2 Mbps. Comparing Figure 9a and Figure 9b, a higher energy conversion efficiency can result in a higher utility for the BS in the complete CSI scenario.

### 7.3. Center User’s and BS’s Utility in Incomplete Channel Information Scenario

We set the center user’s data rate requirement $R_{n}^{min}$ and the center user’s utility (payoffs) as 0.5 and 1, respectively, and the energy conversion efficiency was set to η=0.9 and η=0.7, respectively. Figure 10 presents the BS’s utility (benefit) versus the edge user’s data rate requirement in four groups of incomplete CSI scenarios. The other coefficients were the same as in Figure 9. Figure 10 indicates that, for different groups of incomplete CSI scenarios, with a lower edge user’s data rate requirement $R_{m}^{min}$, the BS would prefer the conventional NOMA transmission mode. Using the proposed cooperative transmission mode, the BS would obtain increased utility with the increase of the edge user’s transmission data rate requirement until a threshold; then, utility would decrease to 0 (i.e., it cannot satisfy the user’s data rate requirement). Under the incomplete CSI scenarios, comparing the four groups, we can observe that high utility for the center user would decrease the BS’s utility, and the energy conversion efficiency would have a great effect on the BS’s utility. This finding should encourage researchers to continue to study and develop chips and circuits technology to improve the energy conversion efficiency to make it possible to meet the diverse needs of wireless power transfer scenarios.

### 7.4. BS’s Utility in Multi-Cooperative-User Sub-6G Networks

The discussion above considers the contract design and transmission power auction in the cooperative transmission process used by the BS to incentivize the center user to participate in the cooperative communication with one edge user. Then, multi-user scenarios with *m* edge users and *n* center users are considered, applying the data rate requirement to BS at the same time. To achieve more utility to accomplish the user’s data requests, the BS needs to design m*n contracts to incentivize the center user to participate in the cooperative transmission.

Considering a scenario with 10 center users and 10 edge users (equal to 10 contract types to one center user), Figure 11a,b present the histogram of the center user’s utility and the BS’s utility, which are obtained from the cooperative transmission in the FD cooperative NOMA and SWIPT transmission mode when the center user accepts the contract from the BS. Each color of histogram refers to the utility obtained from the cooperative transmission from different edge users to the center user. A utility of 0 means that the two matched users cannot satisfy the cooperative transmission demand.

For a single center user, cooperative transmission with different edge users can result in different utilities (payoffs). According to Figure 11a, each center user can obtain its cooperative edge user’s preference list PLm according to amounts of utility in descending order. Meanwhile, to satisfy each edge user’s demand and decrease the power consumption with FD cooperative NOMA and SWIPT transmission, the BS can achieve its cooperative center user’s preference list PLn using the utility (benefit) scores from Figure 11b in descending order. When implementing the same contract, the utility obtained by the center user and the BS are inversely related. More utility being obtained by the BS collecting asymmetric information means that the center user that participates in the cooperative transmission under the incomplete CSI condition obtains less utility, which corresponds to extra transferred energy representation.

The conflict under which the center user and the BS only select their best preference list of cooperative users under the multi-user pairing scenarios results in transmission failure and utility loss. The BS designs contracts for the multi-user incentive mechanism using the GS match theory algorithm to obtain the maximum stable utility.

We set $R_{n}^{min}$ = 2 Mbps, $R_{m}^{min}$ = 2 Mbps, η=0.9. Figure 12 discusses the maximum average utility for the BS versus cooperative transmission user pairs from 4 to 10 in the BS’s coverage areas in 5 transmission and match strategies. For the complete CSI with FD cooperative NOMA and SWIPT transmission, the CU’s utility is 0.5, and this value is 1 for random match and conventional NOMA transmission.

Figure 12 indicates that the utility of the BS obtained from the proposed GS stable match algorithm for FD cooperative NOMA and SWIPT transmission is near to the optimal utility in the analysis with less than 0.2% loss; there is an 8% utility improvement over the random match algorithm, a 3.3% utility improvement over the scenario in which the CU’s utility is 0.5, a 4.9% utility improvement over the scenario in which the CU’s utility is 0.5, and a more than 200% utility improvement over the scenario using conventional NOMA transmission. When the user pairing number is small, the average BS’s utility with the GS match algorithm shows a large fluctuation, which would tend to be stable under different scenarios. Only the average BS utility with conventional NOMA transmission has a major fluctuation owing to the channel utility being zero, meaning that the transmission mode cannot satisfy the matched user pairing’s data rate requirement.

Under the incomplete CSI scenarios, the average BS’s utility is much better than that with the random match strategy and with conventional NOMA transmission strategy. The random match strategy results in a lower average BS utility as some matched user pairs need extra power consumption, meaning that the BS’s utility from that channel decreases. Although the average BS utility in Figure 12 has the lowest average utility, it is still much better than the conventional OMA transmission mode when used to satisfy two cooperative users’ data rate requirements at the same time.

## 8. Conclusions

This paper proposed a contract theory-based incentive mechanism to solve the problem of power allocation and cooperative user pairing with asymmetric information in FD cooperative NOMA and SWIPT transmission communication in a single BS multi-user pairing scenario.

In the design process of a contract, the benefit of the BS is based on the premise of meeting the data rate requirements of the paired edge user and center user and obtaining the corresponding rewards. The BS serving more users at the same time means more benefit, and its cost corresponds to the consumed transmission power. The central user can obtain extra transmission energy from the BS by participating in the FD cooperative NOMA and SWIPT transmission process through asymmetric information. A relaying center user would receive multiple contracts for cooperative communication with different edge users and accept one contract with the maximum payoffs satisfying IC and IR constraints. To guarantee the maximum stable benefit by executing the contracts to meet the users’ requirements, the BS designs the contracts using GS match theory with the preference cooperating list for each edge user.

The simulation results show that the edge user’s data rate in the FD cooperative NOMA and SWIPT transmission is much higher than the conventional NOMA transmission in the low transmission power range; the types of contracts are positively correlated with the channel states of the central user, the BS, and the edge user. The improvement of the energy conversion efficiency can cause the BS to consume less transmission power to ensure cooperative transmission and obtain greater benefits. In the multi-user scenario, the BS adopts the stable matching strategy to design the contract, which can not only mean that the BS obtains more benefits but also leads the center users to obtain expected payoffs.

In the future, the contract theory design of FD cooperative NOMA and SWIPT communication should continue to be studied in millimeter-wave networks with ultra-dense user access (in the special case of vehicular networks) and applied with the graph attention network to strengthen the cooperation between users.

## Figures and Tables

**Figure 1 entropy-23-01161-f001:**
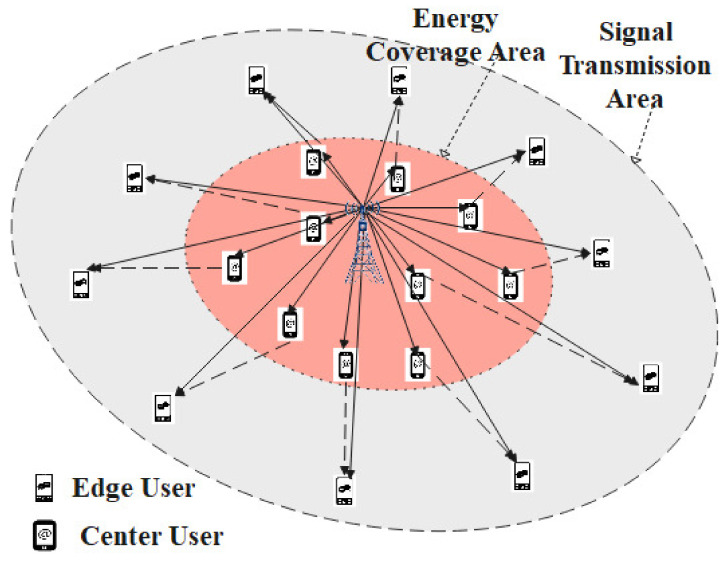
Edge user and center user distribution in sub-6G networks.

**Figure 2 entropy-23-01161-f002:**
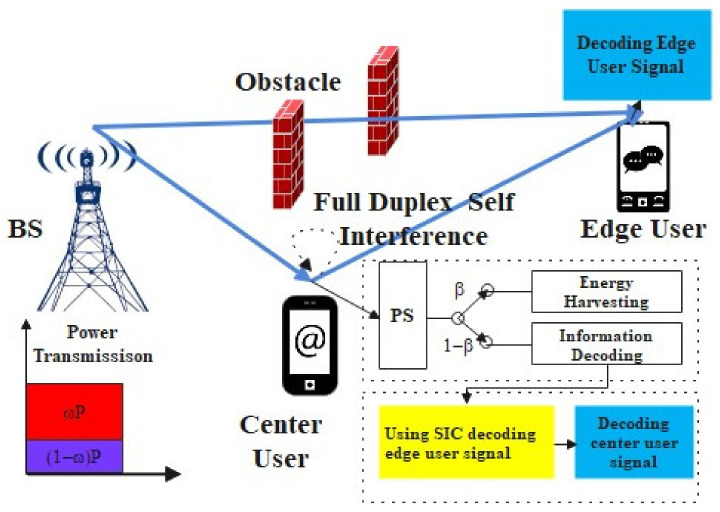
Full duplex cooperative NOMA and SWIPT communication mode.

**Figure 3 entropy-23-01161-f003:**
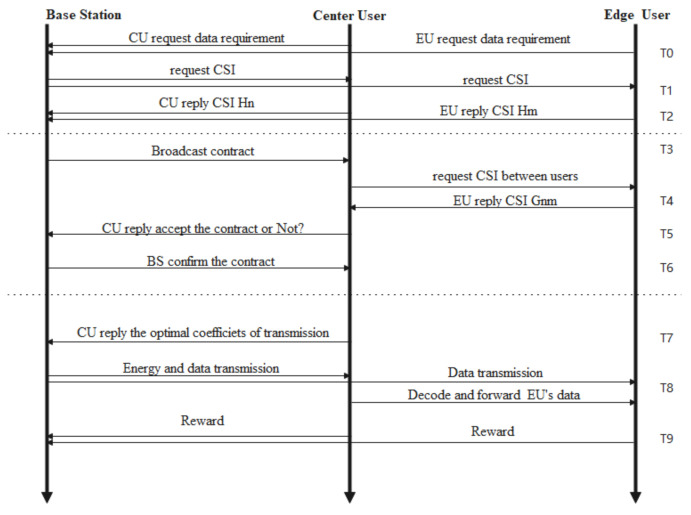
The process diagram of a contract formulation.

**Figure 4 entropy-23-01161-f004:**
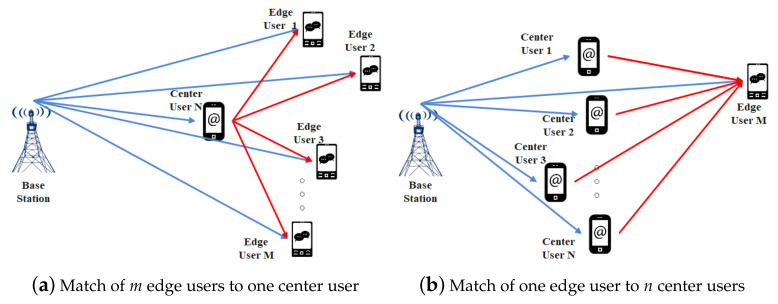
Stable match scenarios.

**Figure 5 entropy-23-01161-f005:**
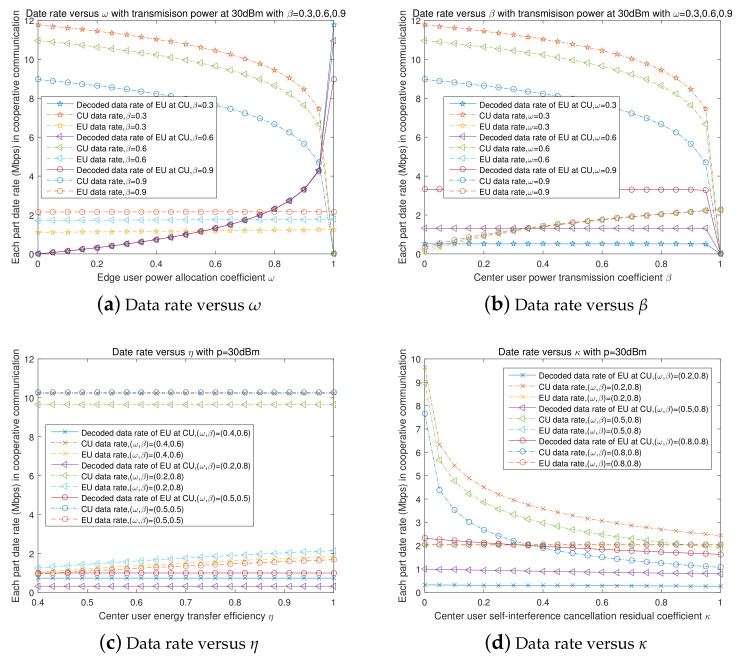
Three decoded data rates versus cooperative network coefficients.

**Figure 6 entropy-23-01161-f006:**
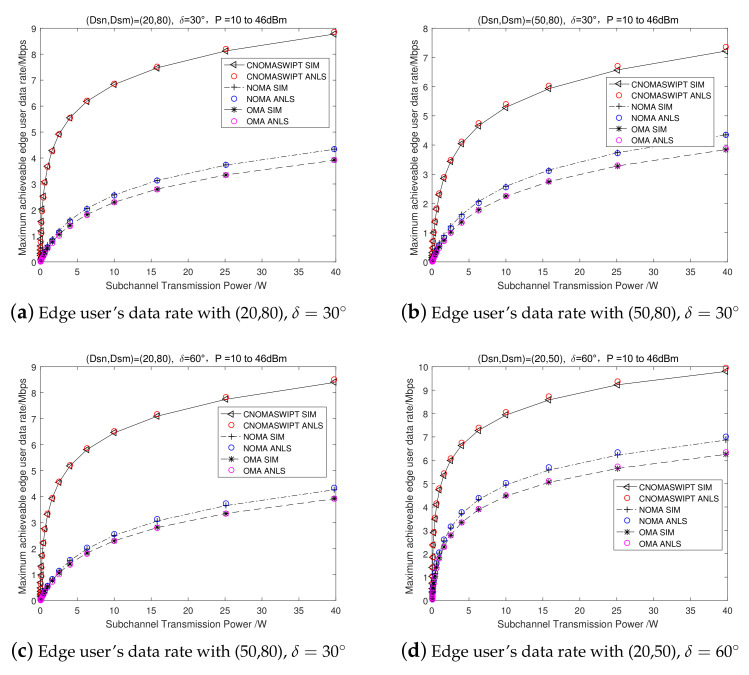
Achievable edge user data rate in Mbps, with $R_{n}^{min}$ = 2 Mbps.

**Figure 7 entropy-23-01161-f007:**
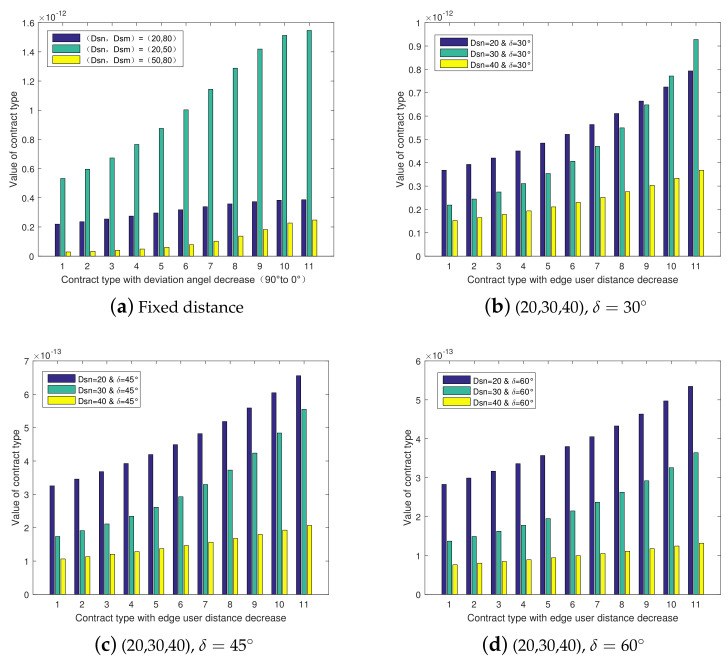
Contract type with edge user distribution.

**Figure 8 entropy-23-01161-f008:**
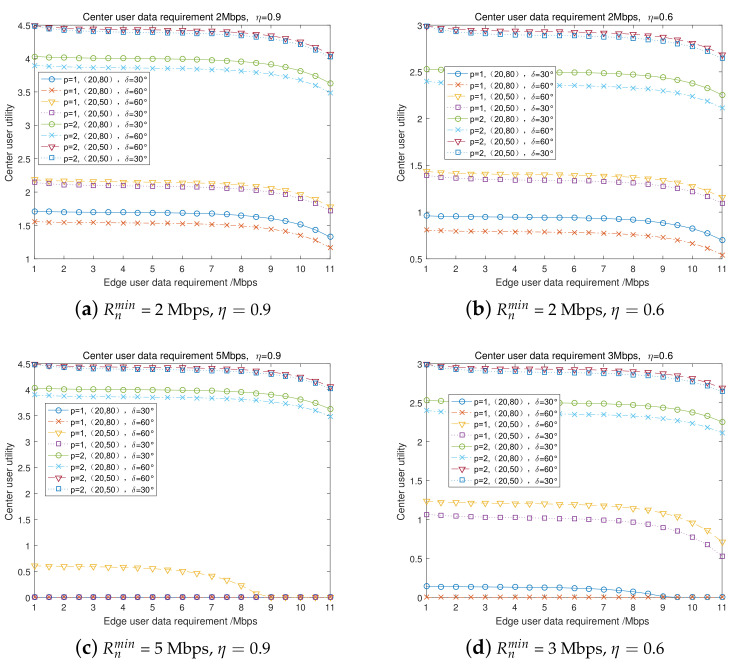
Center user’s utility with fixed transmission power in cooperative communication.

**Figure 9 entropy-23-01161-f009:**
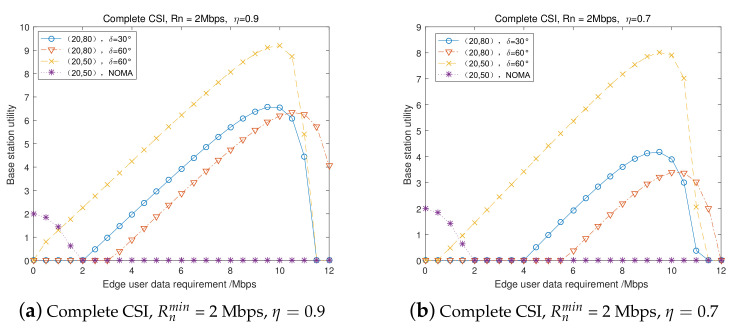
BS’s utility in complete CSI scenarios.

**Figure 10 entropy-23-01161-f010:**
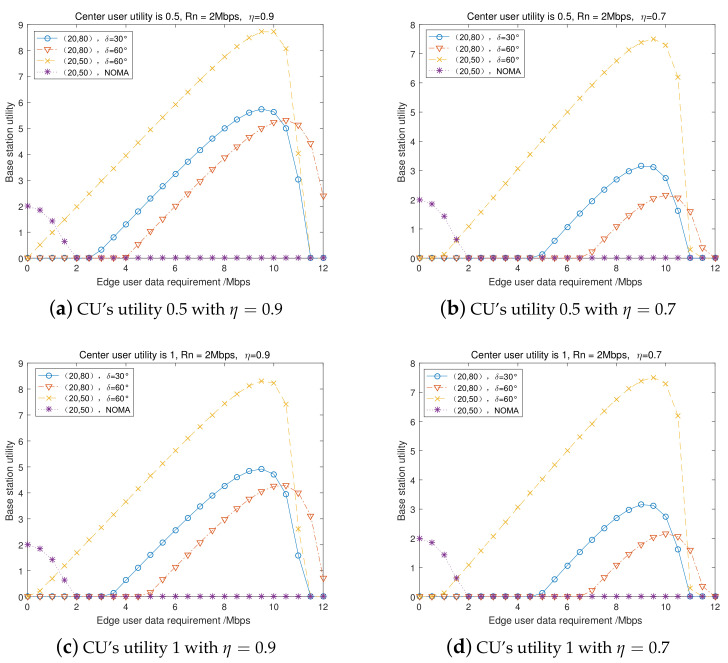
BS’s utility in incomplete CSI scenarios.

**Figure 11 entropy-23-01161-f011:**
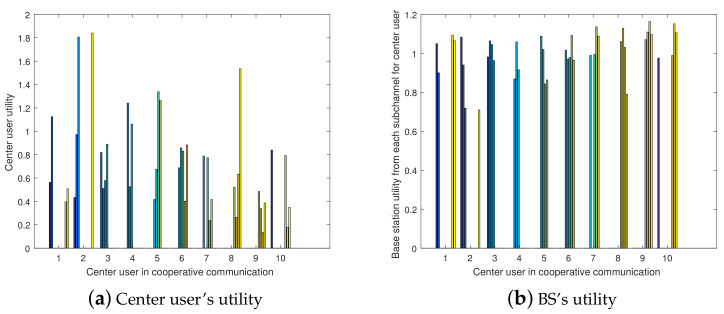
Multi-cooperative users scenario.

**Figure 12 entropy-23-01161-f012:**
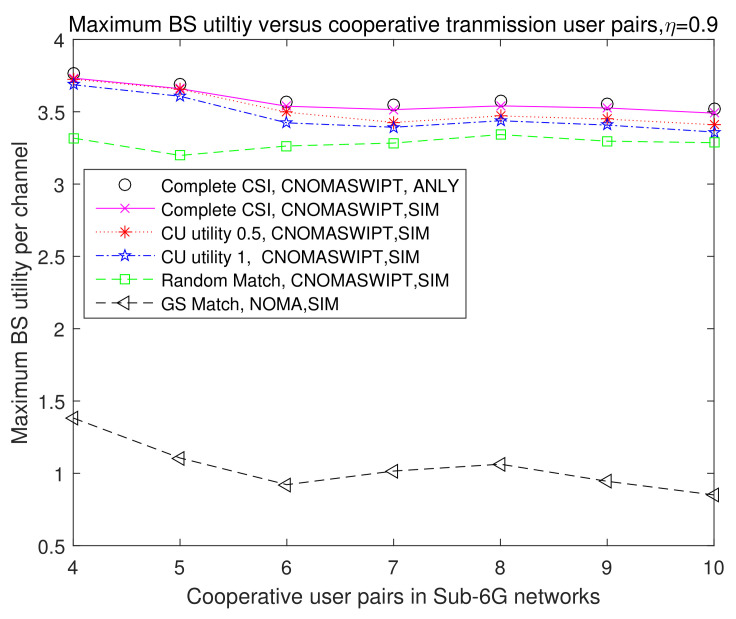
BS’s benefit utility versus the increase of the cooperative user pairing number.

**Table 1 entropy-23-01161-t001:** Simulation parameters in sub-6G networks.

Parameter	Definition	Value
B	Sub-6G network channel bandwidth	10 MHz
$d_{\mathrm{nm}}$	Distance between edge user and center user	50∼ 120 m
$d_{\mathrm{sm}}$	Distance between edge user and BS	50∼100 m
$d_{\mathrm{sn}}$	Distance between center user and BS	20∼50 m
P	Base station transmission power	1∼46 dBm
$p_{c}^{}$	Electricity circuit power consumption	10 dBm
$p_{}^{k}$	Subchannel transmission power	1∼30 dBm
ω	Power allocation coefficient to edge user	0∼1
$\alpha_{m}$	Path loss factor of edge user	4
$\alpha_{n}$	Path loss factor of center user	2
β	Energy transmission coefficient to center user	0∼1
κ	Self-interference cancellation residual coefficient	−80 dB∼0 dB
η	Energy transfer conversion	0.5∼1
δ	Deviation angle from edge user to center user	0∼90${}^{\circ }$
$\sigma^{2}$	Noise power per unit bandwidth	−174 dBm/Hz

## Data Availability

The data presented in this study are available from contracting the corresponding author at denisberg29@gmail.com.

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
