# Peer review of "Contract Theory-Based Incentive Mechanism for Full Duplex Cooperative NOMA with SWIPT Communication Networks"

_entropy, 2021, doi:10.3390/e23091161_

Round 1

Reviewer 1 Report

The authors consider contract-theory-based, stable matching-theory based FD NOMA-SWIPT communication system. The proposed combination of the schemes/scenarios are novel which attracted the reviewer's interest. However, as current shape of the manuscript has poor description, it was hard to follow the details of the paper. Please consider following comments to improve the paper. 

  1. There is no justification for introducing full-duplex system. Please detail why full-duplex system is preferable in the proposed framework. Also, there is no mention regarding full-duplex and SWIPT since the reviewer believes that self-interference cancellation requires certain amount of power which will affect SWIPT -- if it is significant, center users might not want to participate in the cooperation.
  2. Number of edge user (M) = number of relaying centers (N)? While the authors introduced different notations of M and N, it seems that they have to be same. Please make sure whether they can be different or not -- if they are the same, please unify the notation and please justify the scenario where M=N since it seems to be a strong assumption. If they can be different, please detail when they should be same, e.g., for stable matching N=M, while can be different for other cases...
  3. In line 160, it seems that if multiple center users are selected, BS controls its price so that single center user to be selected. However, in Sec. 6, it seems that all the preference level should be defined for the stable matching problem. It is not straightforward how to connect them. First part describes how to assign single center user to single edge user while second part requires all the preferences of the center users (to edge users) and edge users (to center users). Please clarify this point. 
  4. Notations are not defined in the right position. Please define all the new variables in the equations right after that equations, e.g., in (1), subscript 'sn' and 'm' is not defined, 'w' is not described, in line 149, \theta is not defined, etc. This makes all the equations very hard to understand. Please revise all the equations in this regards.
  5. In line 26-28, it seems that 'employee' should not come twice. Please check it again. 
  6. Many typos -- Line 4 'we', line 22 'zhang', line 31 space before '(', line 117 'SIC', line 128 'kth' -- should use italic k, etc. Please revise all the lines to improve readability.
  7. Line 135 -- justification of ignorance of full duplex time delay is missing (please add corresponding citation).

I think the paper has nice contents but current shape prevents me to fully enjoy the paper. Please revise the manuscript accordingly and I'm happy to review in the next round in more detail. Thank you.

Author Response

"Please see the attachment." the same context.

   We have studied the valuable comments on our manuscript“entropy-1354230”from you carefully and do our best to revise the manuscript. We would respond to reviewer’s comments one by one as follows:

   To reviewer 1.

Q1. There is no justification for introducing full-duplex system. Please detail why full-duplex system is preferable in the proposed framework. Also, there is no mention regarding full-duplex and SWIPT since the reviewer believes that self-interference cancellation requires certain amount of power which will affect SWIPT -- if it is significant, center users might not want to participate in the cooperation.

Response:  We are grateful for the suggestion. To be more clearly and in accordance with the reviewer concerns, we have added full duplex (FD) in the abstraction in line 4, and cite four more references [6,7] in line 30, [29,30] in line169, section 2.1. Reference [3-7,29-30] are all study about full duplex and cooperative communication, which indicate that the full duplex transmission moded with decode-and forward is the best choice for the cooperative communication scenarios.

  • Phua, T.T.; Phan, D.; Ha, D.H.; Nguyen, T.N.; Tran, M.; Voznak, M. Nonlinear energy harvesting based power splitting relaying in full-duplexAF and DF relaying networks: system performance analysis.Proceedings of the Estonian Academy of Sciences2020,69.
  • Tseng, M.;  Lee,  T.L.;  Ho,  Y.C.;  Tseng,  D.F.   Distributed space-time block codes with embedded adaptive AAF/DAF elements andopportunistic listening for multihop power line communication networks.International Journal of Communication Systems2017,30, e2950.
  • Wang, S.L.; Wu, T.M. Stochastic geometric performance analyses for the cooperative NOMA with the full-duplex energy harvesting relaying. IEEE Transactions on Vehicular Technology 2019, 68, 4894–4905.
  • Tregancini, A.; Olivo, E.E.B.; Osorio, D.P.M.; De Lima, C.H.; Alves, H. Performance analysis of full-duplex relay-aided NOMA systemsusing partial relay selection. IEEE Transactions on Vehicular Technology 2019, 69, 622–635.

And the power consumption of the SIC in full duplex is a good point to study the problem with full duplex and SWIPT, and this maybe discuss in our future works. The SIC include three parts as :1) the reciever antenna and tranmission antenna’s physical position cancelation, 2) the analog-to-circuit interference cancellation, 3) the digital-tocircuit interference cancelation. The first part-- physical position cancelation is in domain of SIC without power consumption, and the second and third part would consumption which is much smaller than the transmission power for relaying signals from BS to users, and we would neglect it as reference [3-7,29-30].

Also considering the power consumption of the SIC in full duplex mode requires considering the non-linear energy harvesting in SWIPT too, both are not the main studied content in this work. We fouse on the Contract Design with Incentive Mechanism between BS and center user, and to satisfy the edge user and center user’s data requirements while get maximum benefits and payoffs according their contributions in the cooperative communication process.

Q2.Number of edge user (M) = number of relaying centers (N)? While the authors introduced different notations of M and N, it seems that they have to be same. Please make sure whether they can be different or not -- if they are the same, please unify the notation and please justify the scenario where M=N since it seems to be a strong assumption. If they can be different, please detail when they should be same, e.g., for stable matching N=M, while can be different for other cases...

Response:  we revised in line 13 in the abstract added with multi-user pairings scenarios and in the section 2 system model, to emphsize our scenarios as Fig1 that distributed with same numbers edge user m and center user n in cooperative communication one by one with subchannel k, the total number of m, n and k are same(M=N=K), which would be more obviously for the performance comparision. m,n,and k are indicate the user are edge user or center user , and if the communicaiton on the kth subchannel.

We appology for the confusion to you according to our poor english language with not fully and approriate notation following the equation. We revised and added more notations following the equation 1 & equation 5 to help you to understand our model more clearly.

Q3.In line 160, it seems that if multiple center users are selected, BS controls its price so that single center user to be selected. However, in Sec. 6, it seems that all the preference level should be defined for the stable matching problem. It is not straightforward how to connect them. First part describes how to assign single center user to single edge user while second part requires all the preferences of the center users (to edge users) and edge users (to center users). Please clarify this point. 

Response: We appology for the confusion to you that we revised section 3 and section 6, especially the equation 7. The section 3 mainly introduces one contract formulation between BS and one center user (may participate in the cooperation communication with one edge) and in this work the benefit means the reward for the BS to execute the contract and payoffs means the rewards for the center user to cooperative communication with edge user.

Section 6 mainly introduces the M user-pairings scenarios with M edge users and N edge users in the network scenarios, which is a two-side matching problem. And for each edge user, the BS would design a contract between this edge user and each center user as Fig.4b(N contracts). and the BS can get the preference list of this edge user’s cooperative center user by the benefit getting from the communication as equation 7.

And for each center user it would receive M contracts from BS to cooperative communication with M different edge users as Fig.4a, it would calculate the payoffs get from the cooperative communication as equation 12, and getting the relay center user’s preference list by the payoffs.

So totally for the M edge user and N center user scenarios, the BS need to design M*N contracts considering the IC and IR constraints of the center user. With the two-preference list, the propsed GS matching algorith would have lower computation complexity.

Q4.Notations are not defined in the right position. Please define all the new variables in the equations right after that equations, e.g., in (1), subscript 'sn' and 'm' is not defined, 'w' is not described, in line 149, \theta is not defined, etc. This makes all the equations very hard to understand. Please revise all the equations in this regards.

Response:  Thanks for your comments with the detail of our works. And we revised with more notation following the equations.

Added notation in line 162, line 165, line 176 , line 177 and line 208.

Q5.In line 26-28, it seems that 'employee' should not come twice. Please check it again. 

Response: We appology for the confusion. We revised the incentives with incentivize, as

the employer can overcome the asymmetric information and efficiently incentivize (incentives) the employee by offering a contract with performance requirement and corresponding reward.

This is owing to my poor english writing to confuse you.

Q6.Many typos -- Line 4 'we', line 22 'zhang', line 31 space before '(', line 117 'SIC', line 128 'kth' -- should use italic k, etc. Please revise all the lines to improve readability.

Response: Line 4 'we', line 22 'zhang', line 31 space before '(', line 117 'SIC', line 128 'kth' were revised, and we also replaced “we”、“our” with other words, use “FD” replaced “full duplex” and improve other gram and english writing problem. Details in the changes tracking version of the paper.

Q7.Line 135 -- justification of ignorance of full duplex time delay is missing (please add corresponding citation).

Response:  As the answer in Q1, we added two corresponding citations:

  • Wang, S.L.; Wu, T.M. Stochastic geometric performance analyses for the cooperative NOMA with the full-duplex energy harvesting relaying. IEEE Transactions on Vehicular Technology 2019, 68, 4894–4905.
  • Tregancini, A.; Olivo, E.E.B.; Osorio, D.P.M.; De Lima, C.H.; Alves, H. Performance analysis of full-duplex relay-aided NOMA systemsusing partial relay selection. IEEE Transactions on Vehicular Technology 2019, 69, 622–635.

In line 168, revised as: τ is the time delay at relaying center user caused by FD operation, which can be negligible owing to the τ is far smaller than the time slotT[29,30].

In details, even in mmWave communication , one TTI is 0.125ms(in sub-6G or 4Gcommunicaiton one TTI is much larger), the frequence of the full duplex processing chips can work at least of the level of Ghz,which is much smaller than 0.125ms.

Others.

Added a conclusion of the relative work in line 102.

Added an organization of the paper in line 133.

Added two comparison reference in the si,ulation in line 276.

  1. Saito, Y.; Kishiyama, Y.; Benjebbour, A.; Nakamura, T.; Li, A.; Higuchi, K. Non-orthogonal multiple access (NOMA) for cellular future radio access. 2013 IEEE 77th vehicular technology conference (VTC Spring). IEEE, 2013, pp. 1–5.
    34. Fang, F.; Zhang, H.; Cheng, J.; Leung, V.C. Energy-efficient resource allocation for downlink non-orthogonal multiple access network. IEEE Transactions on Communications 2016, 64, 3722–3732.

Added the future work and revised the conclusion.

Revised minor mistakes of writing in section 7.

Reviewer 2 Report

This paper uses matching theory for user pairing and contract theory for power allocation for full duplex cooperative NOMA SWIPT communications.

  • At the end of Introduction, the comparisons with the state-of-the art (such as [8-24] should be summarized.
  • In the simulation results, the baseline algorithms for comparison should be specified as ref xx.
  • In the section system model, decode-and-forward (DAF) relaying is used. In fact, amplify-and-forward (AAF) or adaptive AAF/DAF may be also considered. The author should cite related papers and comment about it such as:

Phua, Tran Tin, et al. "Nonlinear energy harvesting based power splitting relaying in full-duplex AF and DF relaying networks: system performance analysis." Proceedings of the Estonian Academy of Sciences 69.4 (2020). And its references [26][27].

Author Response

"Please see the attachment."  same context.

   We have studied the valuable comments on our manuscript“entropy-1354230”from you carefully and do our best to revise the manuscript. We would respond to reviewer’s comments one by one as follows:

   To reviewer 2.

Q1. at the end of Introduction, the comparisons with the state-of-the art (such as [8-24] should be summarized.

Response:  we added as:

Above on these works, contract theory is used in the wireless communication networks to solve the problem such as content and delivery of D2D networks, cooperative relay selection, NOMA networks and energy transfer/trading networks. This paper would use contract theory for the relaying center user incentive in the FD cooperative NOMA and SWIPT communication networks.

And added the organization at the end of section 1 as:

The organization of this paper as follows: Section 1 gives the introduction of full duplex cooperative NOMA and SWIPT communications , contract theory in wireless communication networks and the contribution of this paper; Section 2 gives the system model; Section 3 gives the design of the contract; Section 4 gives the objective problem; section 5 gives the incentive mechanism design in complete & incomplete channel information scenarios; Section 6 gives the stable matching algorithm; Section 7 gives the discussion in four aspects and Section 8 concludes this work.

Q2.In the simulation results, the baseline algorithms for comparison should be specified as ref xx.

Response:  in the simulation , we discuss the performance of our proposed FD cooperative NOMA and SWIPT communicaiton with different coefficients and compared the conventional NOMA communicaiton as reference [33,34]

Saito, Y.; Kishiyama, Y.; Benjebbour, A.; Nakamura, T.; Li, A.; Higuchi, K. Non-orthogonal multiple access (NOMA) for cellular future radioaccess.  2013 IEEE 77th vehicular technology conference (VTC Spring). IEEE, 2013, pp. 1–5.34.

Fang, F.; Zhang, H.; Cheng, J.; Leung, V.C. Energy-efficient resource allocation for downlink non-orthogonal multiple access network.IEEETransactions on Communications2016,64, 3722–3732.

Q3.In the section system model, decode-and-forward (DAF) relaying is used. In fact, amplify-and-forward (AAF) or adaptive AAF/DAF may be also considered. The author should cite related papers and comment about it such as:

Phua, Tran Tin, et al. "Nonlinear energy harvesting based power splitting relaying in full-duplex AF and DF relaying networks: system performance analysis." Proceedings of the Estonian Academy of Sciences 69.4 (2020). And its references [26][27].

Response: added in line 29 as with two references[6,7]

The performance of FD cooperative relay communication with DF is much better than amplify-and-forward (AF) which is verified by [6,7].

Phua, T.T.; Phan, D.; Ha, D.H.; Nguyen, T.N.; Tran, M.; Voznak, M. Nonlinear energy harvesting based power splitting relaying in full-duplex AF and DF relaying networks: system performance analysis. Proceedings of the Estonian Academy of Sciences 2020, 69.
7. Tseng, S.M.; Lee, T.L.; Ho, Y.C.; Tseng, D.F. Distributed space-time block codes with embedded adaptive AAF/DAF elements and opportunistic listening for multihop power line communication networks. International Journal of Communication Systems 2017, 30, e2950

Also we revised some english wirting mistakes and the changes details in the revised version of paper.

Reviewer 3 Report

The manuscript presents the application of a contract theory-based incentive mechanism under such asymmetric information scenario to incentivize center user to join the cooperative communication to maximize the  Base Stations benefit utility and to guarantee the center user’s expected payoff. 
The proposed cooperative transmission is much better than the conventional Non-Orthogonal Multiple Access transmission and the benefit utility of the  Base Station with the stable match strategy is nearly close to that of the complete channel state information multi-users scenario while the center users get the satisfied expect payoffs.

I find the topic interesting and being worth of investigation and the document is well strucutred, organized, fluidly written, has enough background information, the methodology is clearly explained using correct formulas, the results are adequately presented and support the conclusions.
Although I have the following suggestions:
- Abstract requires structuring such as: problem, motivation, aim, methodology, main results, further impact of those results.
- Keywords should be in alphabetical order.
- The aim should be clarified at the introduction.
- There is a paragraph missing at the introduction explaining how the paper is organized.
- Authors should refrain from using personal pronouns such as "we" and "our" throughout the text and I encourage them to write it in an impersonal form of writing.
- At the conclusion section future lines of research should be disclosed.

Author Response

   We have studied the valuable comments on our manuscript“entropy-1354230”from you carefully and do our best to revise the manuscript. We would respond to reviewer’s comments one by one as follows:

   To reviewer 3

Q1.Abstract requires structuring such as: problem, motivation, aim, methodology, main results, further impact of those results.

Response: we revised as:

Abstract:Cooperative Non-Orthogonal Multiple Access (NOMA) with Simultaneous Wireless Information1and Power Transfer (SWIPT) communication can not only effectively improve the spectrum efficiency and2energy efficiency of the wireless networks but also extend the coverage.

Aim as:

An important design issue is to3incentivize a full duplex (FD) relaying center user to participate in the cooperative process and achieve a4win-win situation to both the Base Station (BS) and the center user. 

Problems as:

Some private information of the5center users are hidden from the BS in the networks.

Methodology as:

 Applying contract theory-based incentive6mechanism under such asymmetric information scenario to incentivize center user to join the cooperative7communication to maximize the BS’s benefit utility and to guarantee the center user’s expected payoff.8Proposing a matching theory-based Gale-Shapley algorithmisproposedto obtain the optimal strategy9with low computation complexity in the multi-user pairing scenario.

Main results as:

Simulation results indicated the network10performance of the proposed FD cooperative NOMA and SWIPT communication is much better than the11conventional NOMA communicationtransmissionand the benefit utility of the BS with the stable match12strategy is nearly close to that of the complete channel state information multi-user pairingsmulti-users13scenario while the center users get the satisfied expect payoffs

Q2.Keywords should be in alphabetical order.

Response: we replaced as:

Contract Theory, Cooperative NOMA, Full Duplex, Incentive Mechanism, SWIPT

Contract15Theory,IncentiveMechanism,CooperativeNOMA,SWIPT,FullDuplex

Q3.The aim should be clarified at the introduction.

Response:  in section 1.2, we revised as:

Motivated by the above findings, the aim of this paper is towedesign110the incentive mechanisms with contract theory under the asymmetric information situation to111incentives the relaying center user to participate in full duplex cooperative NOMA and SWIPT112communication networks that the.TheBS can get the maximum benefit utility while guaran-113teeing the edge user and center user’s transmission rates and the.Thecenter user can get extra114energy transmissiontransferas the payoff to participate in the cooperative communication. The115contributions of this work are fourfold:

Q4.There is a paragraph missing at the introduction explaining how the paper is organized.

Response:  at the end of section 1.2(in line 133), we added organization as:

The organization of this paper as follows: Section 1 gives the introduction of full duplex133cooperative NOMA and SWIPT communications , contract theory in wireless communication134networks and the contribution of this paper; Section 2 gives the system model; Section 3 gives135the design of the contract; Section 4 gives the objective problem; section 5 gives the incentive136mechanism design in complete & incomplete channel information scenarios; Section 6 gives the137stable matching algorithm; Section 7 gives the discussion in four aspects and Section 8 concludes138this work.

Q5.Authors should refrain from using personal pronouns such as "we" and "our" throughout the text and I encourage them to write it in an impersonal form of writing.

Response:  we revised all the “we”,“our” with others, see the details in the changes tracking verion of paper.

Q6.At the conclusion section future lines of research should be disclosed.

Response: we revised the conclusion and added the future lines as:

In  the  future,  the  contract  theory  design  of  FD  cooperative  NOMA  and  SWIPT  com-447munication would continue to be studied in millimeter wave networks with ultra-dense users448access(special case vehicular netwokrs), and applied with graph attention network to strengthen449the cooperation between users.

Other mistakes are revised and the details in the changes tracking version of paper.

Round 2

Reviewer 1 Report

Thank you for the revision that applied all of my comments!

Reviewer 2 Report

No further comments.

This manuscript is a resubmission of an earlier submission. The following is a list of the peer review reports and author responses from that submission.